# Partnership Working among Families, Therapists and Educationalists to Enhance Collaboration Enabling Participation of Children with Intellectual Disabilities

Anu Kinnunen [1,2,*], Annastiina Vesterinen [3], Anu Kippola-Pääkkönen [2] and Maarit Karhula [3,4]

1   Department of Future Health Services, Lapland University of Applied Sciences, 94600 Kemi, Finland
2   Department of Participation and Functional Capacity, Lapland University of Applied Sciences, Jokiväylä 11, 96300 Rovaniemi, Finland; anu.kippola-paakkonen@lapinamk.fi
3   Department of Sustainable Wellbeing, South-Eastern Finland University of Applied Sciences, 50100 Mikkeli, Finland; annastiina.vesterinen@xamk.fi (A.V.); maarit.karhula@kela.fi (M.K.)
4   Research Department, Social Insurance Institution of Finland, 00250 Helsinki, Finland
*   Correspondence: anu.kinnunen@lapinamk.fi; Tel.: +358-4000-487-789

**Abstract:** *Backround*: The collaboration of families, therapists and educationalists has been found to be an important factor in the successful rehabilitation of children with intellectual disabilities. Previous studies have focused on the effects of therapies and the perspectives of parents on collaboration in rehabilitiation. This study aims to describe the facilitators of, and barriers to, collaboration with adults in enhancing the participation of children with intellectual disabilities in education environments. *Methods*: In a qualitative study, the data were collected in 2021–2022 through individual interviews with parents (*n* = 16), focus group interviews (*n* = 17) with education professionals and an online survey tool with open-ended questions from therapists (*n* = 309). The data were analyzed using thematic analysis. *Results*: Two overaching themes were identified: the prerequisites of collaboration and the sharing of expertise in collaboration from the perspective of all the participants, which included the factors facilitating and hindering collaboration. *Conclusions:* In optimizing the benefits of collaboration, all these themes are essential to enhancing functioning and participation. There is a need for the development of participatory practises to improve and intensify collaboration. The development of collaboration and practises should be supported by the managers. Digital solutions should be further explored in order to improve the partnership of adults and children.

**Keywords:** intellectual disabilities; collaboration; rehabilitation

## 1. Introduction

Children's participation and experience of participation have been found to be important for children's and young people's learning, well-being and quality of life [1,2]. The United Nations Convention underlines the right of each person with disabilities to be a full member of society. This is also often an intervention goal for children with disabilities [3]. The World Health Organization (WHO) further defines participation as involvement in a life situation [4]. However, the definition of participation remains unclear. Imms et al. (2017) described participation as consisting of two dimensions: attendance and engagement. Attendance refers to "being there", while engagement refers to involvement in an activity [2]. Legislation frameworks emphasize the importance of attendance—that is, being there; engagement in activities is not emphasized and is often overlooked [5].

The participation of children with intellectual disabilities in daily activities creates opportunities for learning and development. Previous studies have focused primarily on the participation of children with intellectual disabilities (ID) in daily activities. However, there is still a need to discover barriers to and facilitators of their participation in everyday activities [6]. Participation is an amalgamation of two perspectives—the child's

and the caregiver's—as caregivers play a significant role in both accessing and facilitating opportunities for children's participation [7].

According to Arvidsson et al.'s (2013) research, the functioning and participation of a person with an intellectual disability is important for overall well-being [8]. Previous studies have shown that there are more similarities than differences in the participation of children with intellectual disabilities compared to their typically developing peers [9,10]. The ICF (International Classification on Functioning, Disability and Health) of the World Health Organization (WHO) aims to describe how the effects of health and disability affect an individual's life. Functioning is viewed as a dynamic state which consists of the interaction of health status and individual and environmental factors. Examining the interaction relationships of the classification enables the description of everyday life environments in early childhood education, the school and the home, for example. The classification takes into account the importance of participation from the perspective of functional ability [11]. The ICF classification has been found useful when examining the functional ability of children and adolescents with intellectual disabilities and its related environmental factors [12,13]. The ICF classification seeks a synthesis between the different dimensions of an individual's health. It includes all the components of human health. The classification serves as a tool in the collaboration between families and professionals [2].

In rehabilitation settings, how the child participates in their own rehabilitation and the decision-making processes is often examined. Increasing the child's participation in both schools and rehabilitation should be a central principle [14]. Through participation, the child's social skills develop. The working methods and practices of participation should be developed with children, taking into account their age and developmental level [15]. There is a need for the participation-based approach enabling children to learn and develop new skills, while strengthening the family and the child for full participation as members of society [16]. Listening to families and using their expertise helps to find solutions that support the child's everyday life, learning and functioning [17]. Parental and child agency respect leads to a more open and collaborative culture [14]. The family of participation-related constructs also describes processes that influence participation, including preferences, sense of self and activity competences linked to the child's meaningful activities [18].

In promoting the child's participation, the aim focuses on changes in the environment and actions to develop the child's abilities and skills. In such environment-based interventions, therapists, young people and families focus on minimizing environmental barriers and maximizing opportunities [19]. As the child's everyday environment and the activities of people between them are a key part of the child's development and well-being, it is important that the child's parents and other nearby adults receive information about and guidance in how to act with the child [20].

Previous research has focused on the meaning and effects of rehabilitation and therapies in the target group of people with intellectual disabilities on groups with therapeutic symptoms such as Down's syndrome [21,22]. Research has also been conducted on interventions implemented through parents or with parents [23,24]. Greater efforts are needed to support recreational participation in children with ID and consider characteristics to promote participation in leisure activities [9,25].

Previous studies have shown many challenges in active participation at home and school and in leisure time and society for children with disabilities. Parents of children with disabilities have reported a high level of desire to change their children's participation patterns at home and in the community [26,27]. Adults play the key role in strengthening a child's participation in daily life, and the child's experience of participation is formed in these everyday situations [2,14,28]. The concept of family–professional collaboration emphasizes the roles of children and family working together alongside professionals [29]. Underlying this is the idea of partnership, which defines a common role and division of responsibilities in which professionals and parents are equal. The Partnering for Change Model (P4C) is a well-known model for supporting the development of children in need at school, and its purpose is to bring rehabilitation into the school environment. The aim of

the model is to identify children who need support and develop practices in collaboration with parents, teachers and therapists to meet the child's needs. In addition, the goal is to promote children's participation in society, both at school and at home [30,31].

Parental involvement in their own child's affairs is essential as parents possess knowledge of their child, while professionals have special expertise in the situations of similar children through their respective professions. Multiagency entails the involvement and expertise of children and families as the best experts on their own lives. Shared expertise requires a relationship of trust, shared involvement and collaboration [32]. A child's intellectual disability can be a risk factor for both the child's own well-being and that of other family members. Parents may have less free time and be burdened by their use of time, challenges of going to work and financial worries [33–35]. Parents may experience a greater burden and stress because caring for and raising a child often creates more obligations and special requirements. The burden on parents also has a negative impact on the child's well-being. In the case of children with developmental disabilities, it is important to consider and support the parents' ability to cope and the resources and well-being of the entire family [36–38].

Different strategies and collaboration may be needed to facilitate children with ID participation in avoiding social isolation and physical inactivity. These factors are associated with poor health and avoiding longer-term health problems during adulthood [10]. Promoting changes in the environment and family–professional collaboration will optimize the participation of children with disabilities [19,29]. Collaborative intervention, the support of teachers and ecological interventions were identified among the 10 principles by which to guide and deliver interdisciplinary school-based support services for students with disabilities [39]. Finnish children's education in schools and kindergarten is mainly organized with public funding and is highly regulated. Therapies are also mainly organized with public funding, although sometimes the therapy is implemented by therapists from the private sector. There are no therapy services directly integrated into the services of the school and kindergarten, with the exception of government special schools. Thus, there seems to be almost no research data on school and therapy contexts organized in different ways. However, there is a need to better understand how collaborative practices are constructed and what features are influenced in the construction of collaboration in education settings.

The aim of this study is to describe the facilitators of, and barriers to, the collaboration of families, therapists and educationalists in enhancing the functioning and participation of children with intellectual disabilities in education environments. The purpose of this study is to find the factors to improve common participatory practises in the field of education and rehabilitation.

## 2. Materials and Methods

The study was designed as a qualitative study. The data were collected in 2021–2022 through individual interviews with parents (*n* = 16), focus group interviews (*n* = 17) with educational professionals (*n* = 48) and open-ended questions asked of therapists in the Webropol online survey tool (*n* = 309). Charasterics of therapists (*n* = 345) and educational professionals' (*n* = 48) demographics are described in Table 1. There were 245 answers from therapists because some of them had two professions (e.g., physiotherapist and riding therapist). The children of the interviewed parents were aged 1–6 years (*n* = 8), 7–11 (*n* = 6) and 12–16 years (*n* = 2). The parents are located in a geographically extensive area of Finland.

**Table 1.** Participant characteristics.

| Characteristic | *n* | % |
|---|---|---|
| Professional Title of Therapist | 345 | |
| Physiotherapist | 41 | 13 |
| Musical Therapist | 30 | 10 |
| Neuropsychologist | 13 | 4 |
| Psychotherapist | 49 | 16 |
| Speech Therapist | 95 | 31 |
| Riding Therapist | 23 | 7 |
| Occupational Therapist | 94 | 30 |
| Professional Title of Education professionals | 48 | |
| Teacher of Kindergarten | 5 | 10 |
| Special Early Childhood Education Teacher | 9 | 19 |
| Other Staff of Kindergarten | 10 | 21 |
| Teacher of Primary School | 5 | 10 |
| Special Education Teacher | 12 | 25 |
| Other Staff of Primary School | 7 | 15 |

The researcher (A.V) interviewed the parents (16 mothers and 1 father) during the fall of 2021. Fifteen individual interviews and one couple interview (mother and father) were conducted in the Microsoft Teams environment. One interview was conducted via phone. The interviews were recorded in the Teams environment, and the telephone interview with a digital recorder. The interviews lasted 1–1.5 h, and 281 pages of transcribed material were collected. The interviews were conducted as semi-structured theme interviews in which a pre-planned interview frame guided the interviews. The themes were Parent's experience of participating and playing a role in a child's rehabilitation and everyday life as an enabler of rehabilitation.

Early childhood education and school professionals' experiences of rehabilitation and rehabilitation collaboration were conducted by two researchers (A.V. and M.K.) through focus group interviews. The themes in interviews were the forms and development areas of rehabilitation collaboration with therapists and the perceived effects of rehabilitation on the child's functioning and participation in everyday life. A total of 24 people participated in the early childhood education interviews (1 individual interview and 7 focus group interviews), and a total of 22 people participated in the interviews with teaching professionals (1 individual interview and 8 focus group interviews). The duration of the interviews was about one hour. There were 132 pages of transcribed material.

The therapists' Webropol survey was carried out via a public link using a secure connection. The public link enabled several therapists from the same organization to respond to the survey. The response rate cannot be calculated due to the method of data collection.

The data were analyzed via reflexive thematic analysis, which provided a systematic process for coding qualitative material and forming themes from the material [40]. The data were analyzed through inductive orientation. The analysis was guided by a constructionist view of language as a tool for constructing social reality [41]. Braun's and Clarke's (2021) six-step analysis process were applied to analyze and combine three different sets of qualitative data. From step one to step three, all the data (the parents' interviews, early childhood education and school professionals' interviews and therapists' survey data) were analyzed separately, and in steps four to six, the analyses of different data were compared and combined. In the first step, the researcher A.V. became familiar with the interview material, and the researcher A.K. with the open-ended questions of the survey by reading through the material and listening to the interviews. In the second step, the researchers (A.V. and A.K.) started coding each dataset separately and generated initial codes for each dataset. They then grouped the codes and named the code groups. In the third step, preliminary themes were formulated from the code groups. In the fourth step, the entire research team

(A.V., M.K., A.K., A.K.-P.) examined the preliminary themes together and compared the themes in different datasets and the coded material of each dataset. A thematic "map" was generated from this step. In the fifth step, overarching themes were created, and the final themes and their relationships to each other were formed. In the sixth step, the final report was written [40,42].

## 3. Results

The analysis of the integration of different datasets resulted in two overarching themes capturing the prerequisites for collaboration and the features of sharing expertise in daily activities. Results are summarized in Figure 1.

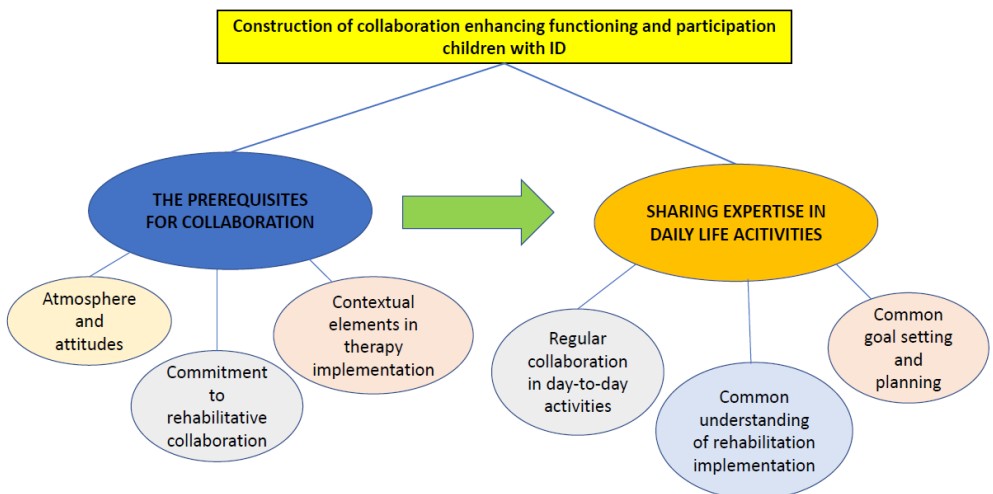

**Figure 1.** Construction of collaboration enhancing functioning and participation of a child with intellectual disabilities (ID).

### 3.1. Prerequisite for Collaboration

Prerequisite for collaboration was constructed through atmosphere and attitudes, commitment to a rehabilitative collaboration and contextual elements in therapy implementation. When these prerequisites for collaboration were met, expertise was thought to be shared to support the child's daily activities.

**Atmosphere and attitudes:** Both the parents and the professionals felt that the open atmosphere, the parents' positive attitude toward collaboration and the professionals and appreciation of each other's skills promoted collaboration. Interest in the work and expertise of others was perceived as meaningful. An understanding of different working methods and cultures of work promoted collaboration and the creation of a multiprofessional atmosphere.

> *Well, the information should change. Regularly. I would miss mutual respect for each other's work. Acceptance that we have different tasks. I think that's enough to get you pretty far (School, interview).*

Regular contact is believed to increase understanding of each other's working methods and culture. However, a lack of time and resources prevents regular interaction. School professionals pointed out that the lack of a common language and understanding between teaching and rehabilitation professionals was an obstacle to an open and respectful atmosphere. This respectful atmosphere is thought to enhance common trust and build a safe environment. These were found to be the cornerstones of collaboration.

> *In a successful collaboration, the observations made individually about the child's concentration, interaction, and learning methods are taken into everyday life and are reinforced. Also, reciprocally: the therapist is told what works in everyday life, and what things could be strengthened in therapy as well (Speech therapist, survey).*

**Commitment to rehabilitative collaboration:** Commitment to rehabilitation collaboration felt to require openness and recognition of the importance of rehabilitation as part of learning and education. Therapists wanted to share their own expertise with early childhood and school professionals. However, they pointed out that the teaching professional's motivation and ability to accept guidance was not always positive, and the implementation of everyday support was therefore inadequate. On the other hand, early childhood education and school professionals felt that therapists did not necessarily value or utilize their expertise.

*It is successful when the kindergarten and school staff think that their actions can influence the child's rehabilitation, and they have the time to do so. It is about commitment (Occupational therapist, survey).*

**Contextual elements in therapy implementation:** According to the professionals, the implementation of therapies as a smooth part of the kindergarten and school environments required the adaptation of therapies to everyday activities. Parents and professionals felt it was important to have a smooth flow of information, to overcome various practical obstacles and challenges related to collaboration and to have enough therapy resources to enable the sharing of expertise.

*That's perhaps what's most important for me, that there would be a group with the therapists, and they have a group on the work phone where they would all be. There should be an assistant from the school or from the preschool if there was one person from all sides who would take care of the child's affairs. So they would be able to work together in a completely different way and not have to be like, well, call this person, call that person, you don't call five different people, and so you can always go and read about it. So, it's like collaboration, that's the most important thing (Parent, interview).*

Suitable facilities that supported the child's activities were seen as important for rehabilitation collaboration. Flexibility in space solutions and schedules were likely to promote the fluent implementation of rehabilitation in teaching settings. This requires therapists to understand the school's everyday life and regular planning and scheduling. Professionals in early childhood education and school raised the challenge of the lack of facilities and their unsuitability for therapy.

*Each actor has his own way of working, so the therapist's instructions are not always welcome. A sense of urgency plagues almost all kindergartens, schools, and housing units, and sometimes the presence of therapists is perceived as pressure. They can't stand adopting new things or believe in the transfer effect. In this case, however, the child's experience in therapy is valuable: the right to come, to succeed, to be a visible and valuable individual, to learn alone without the pressure of a group. It is also noticeable that, for example, communication tools and aids are often little used by older rehabilitators in everyday life, and what is learned in therapy is not reinforced (Occupational Therapist, survey).*

Teachers felt that therapies were important for supporting the child's participation in everyday life. Nevertheless, several teachers reported that the school had decided to outsource therapy due to the lack of facilities, other responsibilities and regular obligations related to collaboration. When the therapies were carried out separately from the everyday life of the school, the therapists and teachers did not become acquainted with each other. This was likely to be followed by a lack of knowledge and understanding of the purposes of individual therapies, and the therapists did not develop an understanding of the child's activities in the everyday life of the school.

*And then there are therapists who don't go to school, so they don't necessarily understand the kind of teaching we do. You have to explain it or if they ask. But they may also have misconceptions about what kind of school we have, and what kind of education the child receives here (School, interview).*

All professionals and parents emphasized that the mutual exchange of information was important. In terms of changing information, it was essential that there was both time and suitable tools for the exchange of information. The challenges of sharing information were the actors' different schedules, therapy taking place outside the kindergarten or school and the lack of common spaces suitable for exchanging information. Professionals felt that information sharing should not merely be the parent's responsibility. On the other hand, teachers hoped for more proactive and active communication from the therapists.

> *Furthermore, I would say that the information moves, the movement of information, that it does not stay in something, is closed to information that does not move. If it is only known to one person on a piece of paper, what good does it do to that child if it doesn't move into everyday life? . . . Part of it comes from openness, and knowledge doesn't move. It's terribly important (Early childhood education, interview).*

School professionals and therapists talked about the challenges in fitting therapies into everyday school life and the curriculum. Regulations and liability issues prevented the implementation of therapies during school days and the participation of school professionals in individual therapies during lessons. Therapists also highlighted the fact that decisions governing the delivery of therapies, e.g., on delivery methods and the amounts of therapy, were bureaucratic, inflexible and too long in duration. Similarly, insufficient therapy resources make it difficult to collaborate. Various restrictions and rules related to school organizations were felt to prevent the collaboration of professionals and the emergence of common ways of working between professionals. The teachers felt that there were unclear questions of responsibility, for example, if the child participated in therapy during a lesson. Therapists also found challenging the fact that they could not become involved in class situations because of the internal rules of the individual school, which prevented the comprehensive planning and implementation of adequate rehabilitation.

### 3.2. Sharing Expertise in Daily Activities

Sharing expertise in daily activities was built through regular collaboration in day-to-day activities, common goal setting and planning and a common understanding of rehabilitation implementation. These were constructed if the prerequisites of collaboration were met.

**Regular collaboration in day-to-day activities:** The sharing of expertise between different professionals and between parents and professionals that supports the participation of the child in day-to-day life was manifested in meetings and discussions between professionals and parents. The challenge was perceived as agreeing on common schedules and thus the lack of common planning. The sharing of expertise took place as joint planning in meetings, as a joint activity of professionals in kindergarten and school and in therapists' guidance situations. The parents participated in the meetings organized in the kindergarten at the school and kept in touch with the professionals using various means of communication. The professionals considered common everyday encounters, discussions and solving challenges to be important.

> *Regular communication would be needed to enable cooperation, participating in fairs once or even twice a year isn't enough. Unfortunately, the everyday life of schools is often so busy that it is impossible to regularly exchange or guide students (Occupational therapist, survey).*

School professionals pointed out that therapists' expertise and therapies were significant for a child's participation and learning at school. The teachers considered the therapists' skills and tips to be very important. They could use them as part of teaching the children and modifying the working environment. The therapists also felt that it was important that their skills were used in the daily life of early childhood education and school.

> *Well, I have, at least from speech therapy. So it's incredibly important, even if the communication folders, building them, and the content that is needed in school, for*

*example. And joint negotiation and planning in which situations they will be used. And feedback from both sides to see how things are going. And whether the child has been supported with the same goals. That's an example of communication (School, interview).*

In everyday collaboration, an important exchange of information about the child's activities and their support occurs. Educational professionals described good experiences with therapists in planning and implementing group situations and lessons. However, the collaboration during authentic everyday activities in child groups or lessons was quite minor. The therapist's participation in authentic everyday activities only in the role of an observer was also perceived as a challenge. The therapist's participation in the classroom was seen as enabling the child to find learning opportunities with the teacher. However, therapists pointed out that it was not always possible to attend lessons. Early childhood education and school professionals hoped that therapists would be more involved in everyday life as part of the work group, and that some of the therapies could be implemented in guided group situations. Early childhood education and school professionals felt that the tangible means and expertise they received from the therapists for everyday activities were important. Therapists also felt that sharing tips and instructions was important.

*I would think that the therapists would bring their own expertise, more know-how to this work. Yes, and it is something when they can focus so much more individually on that one child and their needs... And then also the fact that they may also see the situation from the outside if they come even to that group to see, so they can also give those tips in a way from a different perspective. When you're caught up in everyday life, it may not always see the outside view that, hey, this situation could be quite different (Early childhood education, interview).*

**Common goal setting and planning:** The lack of collaboration was felt to influence the fact that the school professionals and therapists did not form a common vision of the goals supporting the child's everyday participation and work in the same direction. Hectic everyday life and limited opportunities to become acquainted with other actors were perceived as factors weakening collaboration. In the interviews, it was also stated that the COVID pandemic period had had a negative effect on meetings and collaboration in the post-pandemic period. School professionals experienced the lack of contact with therapists and the lack of interaction in everyday life as a challenge. This was mainly influenced by the implementation of the therapies outside the school, as well as the large number and turnover of therapists in the school's everyday life. Encounters often ended up being quick "talks at the door." Meanwhile, therapists felt they were outside the kindergarten and school community when their encounters in everyday life were random.

*It's pretty much exactly that, if the cooperation works, we also know the goals, and we'll be able to implement them in early childhood education and start thinking about them. If there's a challenge, we'll get good instructions from the physiotherapist, this is worth taking into account. It's easy for us to take it forward in everyday life and to look for them in situations where we pay attention. It gets integrated into our everyday life. What we offer here is tied to everyday work. It doesn't happen there, in the everyday doing (Early childhood education, interview).*

Early childhood education and school professionals also found it a challenge that the goals were drawn up separately for therapy, kindergarten and the home, and that they were not known to all the professionals. There were good experiences when an experienced therapist or teacher/early childhood educator had trained an inexperienced therapist. The introduction of an experienced therapist to a new teacher was also perceived as a working practice.

**Common understanding of rehabilitation implementation:** Collaboration in which a common strategy is created helps in finding a common understanding of how rehabilitation is implemented. The therapist's specific expertise benefited the kindergarten and school professionals. The school professionals received tangible practical exercises and support for planning and implementing the teaching from the therapists. Meanwhile, early

childhood education professionals emphasized that the expertise of the therapists gave them new perspectives on their own work, the ability to deal with children holistically and instructions for taking care of their own work ergonomics. The parents pointed out that the guidance of the therapists helped early childhood education and school professionals understand the child and gave them ways to support them. It was felt to be important that things moved from individual therapies to the everyday life of the operating environment.

> *. . . he (therapist) himself said that he also looks at the kindergarten and how the activities are there, that it's not as if he goes there and is there with the boy. So, I thought that was kind of wonderful. If necessary, the other staff are instructed or monitored to see if something needs to be corrected in the activities, or if someone is doing better, if we do things differently with him (Parent, interview).*

Continuous discussion, guidance and collaboration were felt to be essential to ensure that goals and working habits were transferred. Therapists raised the importance of a positive work approach, encouragement and motivation as important factors in collaboration. Encouraging the child to engage in more challenging activities was important, for example, through play and playfulness. This is linked to the strengthening of the child's self-esteem and self-efficacy through successful experiences. Recognizing the child's own resources and needs is important. A shared understanding of the child's needs and rehabilitation goals is important in working together to enable the goals to be achieved.

> *In my opinion, an important aspect of that is also the constant conversation with the therapist about that, or with the therapists about what is being done with that child here in the kindergarten. So that's where the tips come from—hey, we did this, it's going to work well, and this was a meaningful exercise. It also brings us things in common, in a way you are not at the same things, and it also brings assurance that we're doing the right things when we ponder and think about them together (Early childhood education, interview).*

The attachment of rehabilitation to the day-to-day activities of daycare and school was felt to be important. Professionals in early childhood education and teaching considered it important for therapists to participate in school and kindergarten activities by sharing expertise and guiding the staff, as well as by learning about the child's activities in the everyday environment. They felt it was important that the therapists were involved in everyday life, both in observing and modeling activities. Discussion of the findings, feedback and learning from the model strengthened the professionals' skills and understanding of the child's everyday activities, as well as the ability to grasp the child's individual developmental needs and challenges. Therapists also felt it was important that the encounters connected to everyday activities contributed to joint activities for the good of the child. The common "glue" of the activity was described as a genuine interest in working "for the child first".

## 4. Discussion

As a result of the study, collaboration to enhance children's participation with ID is constructed via two main themes: the prerequisites for collaboration and sharing expertise in daily activities.

The prerequisites of a collaborative atmosphere and attitude were seen as important factors which enhanced or hindered collaboration. In other studies, the same factors have also been seen as hindering factors with respect to collaboration and participation [43,44]. The importance of an adult's willingness to collaborate and communicate openly was recognized in this study, as in Nancarrow et al.'s (2013) study [45] study. Willingness was seen as an element building the prerequisite of collaboration, which influences commitment to the rehabilitation process. Commitment enhances day-to-day collaboration by helping to solve and manage the daily situations and needs of a child with a disability. In Anderson et al. (2011), research trust was seen as a key element of collaboration. Common trust

was seen as essential to communication, but also as a factor building commitment to the rehabilitation process [46].

It is important to observe the contextual elements in therapy implementation when examining collaboration. In this study, we found that flexibility in space solutions and schedules promoted better collaboration in day-to-day activities. This required therapists to understand everyday life in the school, as well as regular planning and scheduling. Brewer et al.'s (2012) study [47] also found that successful rehabilitation collaboration was constructed via personal organizational and individual time, energy and resources.

Regular collaboration in day-to-day activities was seen in this study as a key element of participation, enhancing collaboration. Collaboration in everyday life was based on shared information and guided by regular meetings and jointly agreed goals and action plans. A lack of communication or insufficient dialogue between the different actors was also seen as a barrier in other studies, which may have led to ambiguity regarding the various roles and responsibilities in the rehabilitation process [14,46,48]. Means should be developed for practitioners and scholars to better disseminate knowledge in a way that is understood by the profession, its practitioners and its many stakeholders [49].

Based on the results of this study, regular meetings and interactions also emerged as important factors in collaboration which enabled the culture and workings of other professionals to be understood, genuine expertise to be shared and a positive multiprofessional atmosphere to be created. The operating context of therapeutic collaboration is formed by the environment and culture. Sharing expertise in daily activities needs to identify everyone's competence and role in the rehabilitation process. By communicating and sharing expertise, different actors can increase their knowledge of each other and develop a mutual understanding and respect which may facilitate collaboration [46]. According to the results of previous research, collaboration in the daily life of early childhood education and teaching requires a commitment to collaboration and a willingness to learn from each other [50,51].

A common understanding of rehabilitation implementation was seen as crucial in this study for enhancing the participation of a child with disabilities. Taking into account the different perspectives of other professionals and different working habits ensures that common working methods strengthen the child's activity and participation. A recognition of different views and finding a common understanding that promoted the implementation of rehabilitation in everyday life was also seen in Anaby et al.'s (2021) study [39]. Recognizing the expertise of the parents and the child is essential [14,19]. According to this study, therapies for children and adolescents with developmental disabilities should focus even more on collaboration with families. From the perspective of parents, information sharing, being heard and being a partner in all phases of the rehabilitation process are essential [52]. Promoting family involvement in rehabilitation was also proven to be important in Anaby et al.'s (2018) study [19]. The same research showed that the most common implementation strategy identified was training and information exchange, which could facilitate the implementation of the principle of support for teachers and school staff, for example. This further supports emerging capacity building and cost-effective models that aim to empower and increase the competence of school-based personnel through knowledge translation and coaching.

Early childhood education and teaching professionals felt that therapists' support had strengthened their understanding of and skills in activities that supported rehabilitation. However, joint action in rehabilitation was not always successful; it was limited by different operating cultures related to different organizations and how different groups and individuals worked. According to Meuser et al.'s (2022) study [50], the collaboration of professionals (teachers and therapists) in the school environment was valuable for enabling children's participation and also enhanced the sharing of expertise in everyday situations. In practice, the collaboration between therapists and teachers can also be promoted by teachers' greater understanding of the different roles of therapists in the school environment and the means by which therapists can provide value in a classroom [53].

Focusing on collaboration enhancing participation of children with ID through established overarching themes may build effective rehabilitation practices.

*4.1. Strengths and Limitations*

One of the study's strengths is the size of its sample and its national coverage. Geographically speaking, the information producers came from a wide variety of regions; for example, kindergarten and school personnel from different parts of Finland participated in the group interviews. The information producers represented diverse perspectives (parents, therapists, early childhood educators, teachers). The material was extensive, and, as a rule, the interview groups consisted of personnel from a certain kindergarten or school with experience of collaboration with therapists and the implementation of therapies in their unit. It has also been strengthened by a research team with complementary multidisciplinary practical experience and a theoretical understanding of the phenomenon under study and the analysis of multi-perspective qualitative data.

This study has some limitations. The data about the therapist's survey could not be precisely defined due to the method of data collection. Therapists were targeted with the online survey based on the contact information obtained from the Social Insurance Institution of Finland's service provider register, but in addition to this, the bodies representing the therapists were informed about the survey. The parents for the interviews were contacted through a survey in which they consented to being contacted about the interview. It is possible that more parents who were already more active in participating in their child's rehabilitation could have been selected for the interview. All the interviewed parents were highly motivated to participate. This may be due to the method of data collection.

It should also be noted that all those who participated in the interviews with kindergartens or schools worked in cities with more than 50,000 inhabitants, so the interviews do not reveal any special features involved in the collaboration of therapists and early childhood education and teaching staff in smaller towns.

This study had similar limitations to the research of Kinnunen et al. [14]. Different researchers conducted interviews and data analysis, which strengthened the logical implementation of these phases. Thus, this might have had some effects on the interpretations and conclusions. The data management of this study was carried out with the best possible quality and documented accurately, which contributed to the transparency of the study. The internal validity of the study was also increased via the close discussion and cooperation of the examinees throughout the entire research process.

*4.2. Future Directions*

As a result, a "map" of factors to enhance the functioning and participation of a child with ID was constructed. These identified factors should be further researched through the participatory research method. By strengthening the participation of children and families in the planning of services, higher quality and more child- and family-oriented services can be produced. These participatory practices need to be further studied and developed.

Moreover, different strategies are needed in order to implement collaboration—for example, joint training or supervision for parents, therapists and education staff. Managers in working environments also play an essential role, enabling the formation of a common operating culture. Managers should be involved in developing participatory practices. It is also important to educate our students (while studying to be professionals) in participatory practices when working with children, parents and other professionals. Encounters between different actors are important during studies; they prepare for future collaboration as professionals.

Digitalization can be a key factor in improving the collaboration of adults and enhancing the participation of children with ID. Digital tools can help us to become closer to the daily lives of children and their families and understand their needs in a new way. Collaboration and interaction between different actors can also become more agile.

## 5. Conclusions

This study's results focus on collaboration enhancing the participation of a child with intellectual disabilities. The collaboration between parents, therapists, early childhood education and teaching professionals related to rehabilitation was constructed via prerequisites of collaboration such as atmosphere, attitudes and conceptual framework in the implementation of therapy. These prerequisites for collaboration are linked to the sharing of expertise that supports the child's everyday participation. Sharing expertise in daily activities in collaboration was built by identifying expertise, regular collaboration in day-to-day activities and common goal setting and planning. Through these elements of collaboration, participatory daily life can be achieved.

**Author Contributions:** Conceptualization, A.K., A.V., A.K.-P. and M.K.; methodology, A.K., A.V. and M.K.; writing—original draft preparation, A.K., A.V., A.K.-P. and M.K.; writing—review and editing, A.K., A.V., A.K.-P. and M.K. All authors have read and agreed to the published version of the manuscript.

**Funding:** This research was funded by Social Insurance Institute of Finland. Funding number: Kela 25/26/2020.

**Institutional Review Board Statement:** The study was conducted according to the guidelines of the Declaration of Helsinki, and research approval was obtained from The Social Insurance Institution's research ethics committee, no 15/500/2020.

**Informed Consent Statement:** Informed consent was obtained from all subjects involved in the study.

**Data Availability Statement:** The data presented in this study are not available due to participant privacy.

**Conflicts of Interest:** The authors declare no conflict of interest.

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
