# Peer review of "Partnership Working among Families, Therapists and Educationalists to Enhance Collaboration Enabling Participation of Children with Intellectual Disabilities"

_disabilities, doi:10.3390/disabilities3030026_

Round 1
Reviewer 1 Report
Dear authors,
Thank you very much for this interesting contribution.
The paper is about adults working together to improve the functioning and participation of children with intellectual disabilities (as the title puts it) The attitudes and experiences of the people involved are identified through semi-structured interviews, focus groups, and open-ended questionnaires with therapists, teachers, and parents of children with intellectual disabilities. The introduction clearly and comprehensibly identifies the research gap that the described research aims to fill. This helps build a better understanding of the stakeholders to consider in care.
The paper is well described and understandable, and the sources are current and sufficient. There are some spelling errors, so the text should be checked again. The method and results are well presented. It is clear what the attitudes of the interviewees are. The discussion is well written and relates to the results and the state of the research. There are no other comments except for the following point:
Please check the spelling of the text: e.g.: line 71 - if this is a name, please label it clearly and line 134 professionals.
Author Response
Thank you for your revision and suggestions to improve our manuscript. We have now carefully read and corrected the spelling mistakes in the manuscript. All the changes are marked in red colour.

Reviewer 2 Report
In a qualitative study, the authors report on semi-structured interviews, focus groups, and open-ended questionnaires with therapists, teachers, and parents of children with intellectual disabilities. Qualitative studies complement already existing results from hypothesis-driven objective studies (e.g., with validated questionnaires) by providing in-depth insights into the motivation and attitudes of the people involved. In this sense, the presented study contributes important aspects to the topic and can facilitate both the personal attitudes of the involved groups of persons and planning to improve the care of intellectually disabled children.
In my view, the following aspects of the manuscript could be improved or supplemented:
1. in the present qualitative research, personal experiences and subjective attitudes of the actors were recorded. This could be more positive ("has been my experience") or more negative ("has not been realized"). However, in the manuscript some of the findings are presented not as hypotheses but, as facts. I think it is important to distinguish provable facts and collected subjective assessments textually throughout and clearly. Example: "the collaboration of adults enhances the functioning and participation of children" would be better: "is believed/thought/expected/..." to enhance
2. the experiences of the interviewed actors certainly depend on the context of their activity (private or state kindergartens and schools, special kindergartens and schools for disabled, integrated institutions for disabled and normal children, therapists in private practice or integrated in the institutions, regular involvement of parent meetings institutionalized or not, separate funding of parent counseling and interprofessional exchange,...). Are there any data and differences on this within the Finnish clientele studied? This information would be important at least for an international readership.
there are a few typos in the text: page 3, line 134: described; page 4, line 175/76: repond
there are a few typos in the text: page 3, line 134: described; page 4, line 175/76: repond
Author Response
Thank you for your valuable comments. We have now corrected our manuscript based on your suggestions. All changes are in red colour in the manuscript.
- We have reformulated our title of the manuscript and rephrased our findings in some parts of the text.
- Thank you for your valuable comment. We have now tried to describe the Finnish context on education system and therapies. In Finland our education system and therapies are mainly organized with public funding, so there is not so much data about the differently organized collaborations.
- Spelling mistakes have been corrected.

Reviewer 3 Report
My recommendation would be to accept the paper in its current form, as I identify only minor proofreading errors.
If I had to make one suggestion for improvement, it would be to find more systematic methods to present the results of interviews and focus groups.
Overall english proofreading is needed before publication. No major issues were detected.
Author Response
Thank you for your revision and valuable comments. We have now proofread our manuscript and marked all the changes in red colour.
